# Unpacking the Public Health Triad of Social Inequality in Health, Health Literacy, and Quality of Life—A Scoping Review of Research Characteristics

**DOI:** 10.3390/ijerph21010036

**Published:** 2023-12-27

**Authors:** Heidi Holmen, Tone Flølo, Christine Tørris, Borghild Løyland, Kari Almendingen, Ann Kristin Bjørnnes, Elena Albertini Früh, Ellen Karine Grov, Sølvi Helseth, Lisbeth Gravdal Kvarme, Rosah Malambo, Nina Misvær, Anurajee Rasalingam, Kirsti Riiser, Ida Hellum Sandbekken, Ana Carla Schippert, Bente Sparboe-Nilsen, Turid Kristin Bigum Sundar, Torill Sæterstrand, Inger Utne, Lisbeth Valla, Anette Winger, Astrid Torbjørnsen

**Affiliations:** 1Department of Nursing and Health Promotion, Faculty of Health Sciences, Oslo Metropolitan University, 0130 Oslo, Norway; tonenyga@oslomet.no (T.F.); ctorris@oslomet.no (C.T.); borglo@oslomet.no (B.L.); kalmendi@oslomet.no (K.A.); anki@oslomet.no (A.K.B.); elenaaf@oslomet.no (E.A.F.); ellgro@oslomet.no (E.K.G.); solvi@oslomet.no (S.H.); liskva@oslomet.no (L.G.K.); rosahm@oslomet.no (R.M.); nimi@oslomet.no (N.M.); anrasa@oslomet.no (A.R.); kiri@oslomet.no (K.R.); idahan@oslomet.no (I.H.S.); anacarla@oslomet.no (A.C.S.); bnilsen@oslomet.no (B.S.-N.); turidkri@oslomet.no (T.K.B.S.); torillma@oslomet.no (T.S.); inger@oslomet.no (I.U.); lisval@oslomet.no (L.V.); anwin@oslomet.no (A.W.); astridto@oslomet.no (A.T.); 2Intervention Centre, Oslo University Hospital, 4950 Oslo, Norway; 3Department of Surgery, Voss Hospital, Haukeland University Hospital, 5704 Voss, Norway; 4Department of Rehabilitation Science and Health Technology, Faculty of Health Sciences, Oslo Metropolitan University, 0130 Oslo, Norway; 5Faculty of Medicine and Health, Örebro University, 701 82 Örebro, Sweden; 6Regional Centre for Child and Adolescent Mental Health, Eastern and Southern Norway (RBUP), 0484 Oslo, Norway

**Keywords:** social inequality in health, health literacy, quality of life, health promotion, public health

## Abstract

Social inequalities in health, health literacy, and quality of life serve as distinct public health indicators, but it remains unclear how and to what extent they are applied and combined in the literature. Thus, the characteristics of the research have yet to be established, and we aim to identify and describe the characteristics of research that intersects social inequality in health, health literacy, and quality of life. We conducted a scoping review with systematic searches in ten databases. Studies applying any design in any population were eligible if social inequality in health, health literacy, and quality of life were combined. Citations were independently screened using Covidence. The search yielded 4111 citations, with 73 eligible reports. The reviewed research was mostly quantitative and aimed at patient populations in a community setting, with a scarcity of reports specifically defining and assessing social inequality in health, health literacy, and quality of life, and with only 2/73 citations providing a definition for all three. The published research combining social inequality in health, health literacy, and quality of life is heterogeneous regarding research designs, populations, contexts, and geography, where social inequality appears as a contextualizing variable.

## 1. Background

Social inequality in health, health literacy, and quality of life are all important measures of health and are related to each other [1,2,3,4,5]. Thus, numerous policy documents have emphasized the value of each of these concepts, though often as standalone objectives. As major public health problems tend to be unequally distributed within populations, increased efforts towards improving health will, to a large extent, be connected to equalization in health, as well as to improving quality of life and health literacy [1,2]. Against the backdrop of recognized public health challenges, there is a need to consolidate the existing literature on the interplay of social inequality with health, health literacy, and quality of life. These three key indicators, while individually significant, collectively offer a comprehensive lens through which to address public health issues, in line with the United Nations’ Sustainability Goals [6]. By mapping current research and identifying gaps, we contribute to a strategic approach in public health practice and research, aligning with global objectives for sustainable health.

Social inequalities in health can be defined as “(…) health disparities, within and between countries, that are judged to be unfair, unjust, avoidable, and unnecessary (meaning: are neither inevitable nor unremediable) and that systematically burden populations rendered vulnerable by underlying social structures and political, economic, and legal institutions” [7]. Consequently, inequalities in health can occur along socioeconomic, political, ethnic, and cultural axes [5,8]. Often, within those groups suffering from social inequality, levels of health literacy tend to be lower [3,4,9]. Because health literacy may be defined as “the cognitive and social skills which determine the motivation and ability of individuals to gain access to, understand and use information in ways which promote and maintain good health” [10], levels of health literacy can have consequences for health, but the picture is complex [3,11]. Health literacy both mediates and moderates self-assessed health and has the potential to predict health [11]. Similarly, parental health literacy may appear as a family trait linked to health inequality, associated with children’s health outcomes and the development of health inequalities from birth [12]. Health literacy may hold the potential of being an intervenable and direct determinant of health [11]. For example, interventions targeting high-risk groups have been beneficial among a general population [13], among people with noncommunicable diseases in low- to middle-income countries [14], and among adults with chronic conditions [15]. However, the heterogeneity among the study methods is considerable, and the often-applied Western perspectives on health literacy can impose negative consequences on both research and practice [16]. Examples of such could include the exclusion of groups from research based on their social, ethnic, cultural, or geographical backgrounds, or the use of measures and forms of health literacy that have not been adapted or validated for each specific population, resulting in biased conclusions [16]. As a result, populations may receive unequal access to health services.

Quality of life also tends to be unequally distributed among socioeconomically diverse populations [5]. As a broad term, quality of life includes well-being related to different aspects of existence. A commonly applied definition is “(…) *an individual’s perception of their position in life in the context of the culture and value systems in which they live and in relation to their goals, expectations, standards, and concerns*” [17]. Specific to health, health-related quality of life is commonly defined as *“(…) referring to the health aspects of quality of life, generally considered to reflect the impact of disease and treatment on disability and daily functioning; it has also been considered to reflect the impact of perceived health on an individual’s ability to live a fulfilling life*” [18]. In a recent large-population study in South Korea, socioeconomic transition predicted changes in health-related quality of life [19], supporting the formerly observed negative social gradient associated with poorer quality of life. Similar socioeconomic gradients have been observed elsewhere, for instance, as being related to oral health-related quality of life [20] and children with asthma, epilepsy, type 1 diabetes, or chronic kidney disease, all showing lower socioeconomic status to be associated with poorer quality of life [21]. Another systematic review focusing on people with chronic illnesses showed that socioeconomically disadvantaged groups had increased levels of health impairments and lower health-related quality of life once their health was impaired [9].

There are also dual relationships between the concepts of health literacy and quality of life. Although the definition above, as proposed by Nutbeam [10], focuses on the relationship with health, another definition of health literacy implies an association with quality of life: “*the wide range of skills, and competencies that people develop to seek out, comprehend, evaluate, and use health information and concepts to make informed choices, reduce health risks and increase quality of life*” [22]. Similarly, the WHO has acknowledged well-being in their recently updated definition of health literacy [23], supporting this dual relationship. In a systematic review of studies from 1970 to 2018, a moderate correlation between health literacy and quality of life was demonstrated [24]. Furthermore, a significant association between higher levels of health literacy and higher quality of life was observed in patients with chronic kidney disease [25]. However, both of these studies require more research to support their findings of moderate associations.

Overall, it remains unclear how and to what extent the three concepts of social inequality in health, health literacy, and quality of life are interrelated and how this is reflected in the literature. At the same time, the characteristics of the research have yet to be established [1,26,27]. Consequently, a scoping review can provide a systematization of the existing knowledge about these concepts in combination that may aid in developing new research questions on the mutual relationship among them in different populations, patient groups, and contexts. Furthermore, such a systematization may provide support for policymakers in their efforts, for example, to reach the United Nations Sustainability Goals.

Thus, this scoping review aims to identify and describe the characteristics of research that intersects social inequality in health, health literacy, and quality of life. The following primary research questions guided this study:What are the characteristics of the studies that use the three concepts of social inequality in health, health literacy, and quality of life together?How are the three concepts presented in the research?Which definitions and measurements are used for the three concepts?

## 2. Materials and Methods

### Design

A protocol for our scoping review was developed a priori [28]. According to the scoping review framework proposed by Arksey and O’Malley [29], and extended by Levac et al. [30], the work proceeded in six steps: Step 1: Identifying the research question; Step 2: Identifying relevant studies; Step 3: Data selection; Step 4: Charting the data; Step 5: Collating, summarizing, and reporting the results; and Step 6: Consultation exercise. The results are reported according to the Preferred Reporting Items for Systematic Reviews and Meta-Analyses Extension for Scoping Reviews (PRISMA-ScR) Statement [31]. There were no deviations from the initial protocol.

Step 1: Identifying the research question

The multidisciplinary research group that conducted the current scoping review has worked or conducted research within the fields of social inequality in health, health literacy, and quality of life from a life course perspective. The participating authors hold positions within research, education, and faculty research leadership at academic levels from Ph.D. candidates to professors. Within the research group, social inequality in health, health literacy, and quality of life have been discussed and investigated in various independent projects. Consequently, we have explored the possible connections between these concepts and the possibility that these mutual connections may be neglected more or less frequently in available publications and strategic documents. Conducting a full systematic review was regarded as premature, as the field has not been exhaustively studied previously, and it remains necessary to consolidate the existing literature about the combined study of social inequality in health, health literacy, and quality of life. To better inform future systematic reviews, meta-syntheses or meta-analyses, a scoping review was seen as crucial to investigating the characteristics of this field to ensure that future reviews extract and synthesize the relevant variables in the required contexts and populations [29]. Thus, the overall research question in our proposed scoping review is as follows: “When social inequality in health, health literacy, and quality of life appear together in a publication, how are they used, presented and/or discussed?” To translate the research question into a searchable string for databases, we applied the population, concept, and context (PCC) tool (Table 1) [32].

Step 2: Identifying the relevant studies

**Eligibility criteria.** Eligible citations had to apply the concepts of social inequality in health, health literacy, and quality of life combined, regardless of population or context (Table 1). To be considered in combination, the three terms had to be logically associated in the report. Potentially synonymous terms were discussed, and the final selection included (1) social inequality in health: social determinants of health, health status disparities, socioeconomic status, health service accessibility, and differences in sociodemographic characteristics among groups; (2) health literacy: health competence, medical literacy, and understanding (of health), in addition to specific instruments such as the health literacy questionnaire (HLQ) [33] and the European Health Literacy Survey Questionnaire (HLS) [34]; and (3) quality of life: life satisfaction, personal satisfaction, or specific instruments such as SF-36 [35], RAND [35], EQ-5D [36], and PedsQL [37]. Citations were included irrespective of publication type, year of publication, or language. Systematic reviews were also considered as eligible.

**Information sources and search**. The searches were built by HH, AT, CT, TF, and BL under the guidance of a university librarian until a satisfactory search string in Medline was agreed upon and translated into the other databases. The final searches were conducted by HH and AT in Medline, Excerpta Medica Database (EMBASE), Cinahl, the Allied and Complementary Medicine Database (AMED), PsychInfo, Scopus, Cochrane, Epistemonikos, the Social Sciences Citation Index (SSCI), and Applied Social Sciences Index & Abstracts (ASSIA). The chosen databases reflect the broad scope of our research question, including both health and social perspectives, and all search strings are found in the Appendix A.

Step 3: Data selection

Data screening was assisted by the systematic review software Covidence by Veritas Health Innovation [38]. Deduplication was conducted twice, first in Endnote^TM^ before uploading references to Covidence and then repeated in Covidence. Screening of titles and abstracts, followed by a full-text review, was conducted in Covidence. The first initial 500 citations were screened independently by five members of the group (HH, AT, CT, TF, and BL) to assess the relevance of the citation according to our inclusion criteria and to check for any possible need for revisions to the criteria. At this piloting stage, publications that did not comply with the aims of our research were subject to discussion among the five reviewers before exclusion. Similarly, to reduce variability in assessment, the remaining citations were screened independently by two group members, one of whom was always either HH, AT, CT, TF, or BL. The first and last authors of our scoping review were responsible for resolving any conflicts at any stage of the review.

Step 4: Charting the data

**Data extraction.** The final citations were subject to extraction through a prespecified charting form developed by HH, AT, CT, TF, and BL. The entire text of the included citations was regarded as material for analysis because the concepts were applied differently from paper to paper. The extraction form was programmed in Covidence and covered publication characteristics such as author, year, country, context, population, and methods. The data on the definition and use of the three terms included (1) the relevant definition for all three concepts as written in the paper, if any, (2) any use (and definitions) of interchangeable concepts to either of the three, (3) a note on where in the paper the concepts were applied, (4) a note on how the three concepts were combined, (5) a note on what, if anything was seen as the main concept and whether there was a hierarchical relationship between the concepts, and (6) a summary note on the included and extracted report.

Following data extraction, the context was categorized according to three levels of service, community-based, regional- or district-level based, or national or international-level based, as inspired by a previously applied categorization [39]. Populations were captured by demographic descriptions in the included reports, divided first by age groups, and then further categorized into the following groups: patients (subdivided into diagnostic groups), immigrants/minorities, Inpovered populations, general populations of entire nations, or others (including students and employees). The data on the methodology of the reviewed citations included whether it was qualitative, quantitative (with further details on its design, as classified by the authors of the citation), multi/mixed method, or reviews. Reports without a specific design were labelled as orientational reports, including statements, editorials, and consensus papers. For each of the three key concepts, reports were categorized on whether a definition was explicitly stated (yes/no), and whether the concepts were applied in the introduction, method, results, or Section 4 of the citation, whenever possible. In the category in which the definitions were provided, these were cross-referenced by an analysis using NVivo [40] in the next step. Further, we investigated how the three concepts were combined, that is, if there was one main variable of interest and if there was a hierarchical relationship among the three. The charting form was tested on the first 10% of the included full-text papers and found to be satisfactory. All the authors took part in the data extraction in predefined pairs.

Step 5: Collating, summarizing, and reporting the results

**Analysis**. The data extraction form was subject to descriptive analysis using Microsoft Excel, version 2208. To investigate any overlap between the included primary studies of the reviews and our included primary studies, a cross-table of studies was developed, revealing less than a 3% overlap. According to scoping review guidelines [32], we did not conduct any methodological appraisals.

Step 6: Consultation exercise

To generate discussion and feedback on the process and preliminary findings, we involved researchers at all levels from PhD candidates to professors holding positions within research, education, and faculty leadership. Arenas for input included regular research group meetings at Oslo Metropolitan University and a national conference for health literacy research, Helinor 2022.

## 3. Results

### 3.1. Search Results

From the 4111 initial records, 300 citations were screened for eligibility in full text, of which 73 citations were included in the full review (Figure 1). An overview of the characteristics of these fully reviewed citations is provided in Appendix A.

### 3.2. Characteristics of the Included Citations

The citations were published from 1999 to 2022, with 44 reports (60%) published before 2019 and 29 (40%) thereafter. The included citations comprised 70 full papers, which included protocols (n = 8), reviews (n = 5), dissertations (n = 3), and orientational reports (statements, editorials, and consensus papers) (n = 7). Three citations were conference abstracts. Among the 61 primary studies included, most were conducted in North America (n = 33), 11 were carried out in Asia, nine in Europe, two in South America, two in Africa, and two in Australia. One study was a joint study undertaken in the United States and Australia [41], and one study was undertaken in Italy, Netherlands, Spain, France, and the UK [42] (Figure 2). Among the 61 primary studies, quantitative methods were applied in most of the studies (n = 50), while qualitative methods were applied in five studies, mixed or multi-methods were used in five studies, and one used a case-study method. The rest included five reviews and seven orientational reports. Cross-sectional designs accounted for 47% (34/73) of all included studies in this review. Eleven citations reported on research with an international or national context, 21 studies were at a regional or district level, and the majority (n = 41) were in a community context. Details on all included studies can be found in the Appendix A.

According to the assessed populations, the 73 included citations were categorized into five groups of either patients with varying diseases (n = 34), Inpovered populations (n = 17), immigrants or other minorities (n = 8), general populations of entire nations (n = 7), or others (n = 7) (Figure 3). Eleven citations primarily concerning patients also belonged to either the inpovered populations (n = 9) or the immigrant/minority group (n = 2), and one study could be categorized into both the inpovered populations and the immigrant/minority groups. Therefore, these 12 studies were recorded twice each. Of the citations, three focused on mental health, including one on psychological distress among elders in rural Bangladesh [43], one on mental illness among patients with cancer in disadvantaged communities [44], and one on the prevalence of depression and anxiety in low-income or poor areas in Malaysia [45]. Children were the primary study population in one [46] and were included through samples of families in six studies [47,48,49,50,51].

The contextual level in association with the research design of the 73 publications is shown together in Figure 4. There appears to be an even distribution of the reported study designs among the three context levels.

### 3.3. Presence, Definitions, and Measurements of the Three Concepts

All the included reports used the prespecified concepts in their text, but they were given different priorities. All three concepts had a joint focus in six reports, while the primary focus was social inequality in health in 12 reports, health literacy in 29 reports, and quality of life in 13 reports. Social inequality in health and health literacy had a joint focus in three reports, health literacy and quality of life had a joint focus in eight reports, and social inequality in health and quality of life were the focus in two reports.

The included reports varied as to whether definitions for the three concepts studied were provided in the publications (Table 2a,b). For the 55 quantitative studies (including those with mixed methods), measurements to assess concepts of interest also varied among the included reports (Table 2a). We did not investigate the way the concepts were assessed in the qualitative studies, reviews, or orientational reports.

The studies rarely contained definitions or only one measure to assess social inequality in health; instead, the authors used social inequality in health to contextualize their study, referring to a framework or to their understanding of the concept related to their objectives. Similarly, a combination of variables was often claimed to represent elements of the social inequality of health. These variables included income, education, gender, years/level of education, type of residence, employment status, insurance and access to health care, family structure, and marital status, among others. Among the included reports, 11/73 provided a clear definition of social inequality in health according to their contexts. Variables to assess social inequality in health were assessed in 34/55 of the primary studies using quantitative or mixed methods. The concept of health literacy was defined in 41/73 reports and operationalized in 50/55 primary studies with quantitative or mixed methods. A definition of quality of life was present in 12/73 reports, and measurements of quality of life were identified in 46/55 studies through the use of quantitative or mixed methods. Definitions for all three concepts were identified in only three of the included reports [84,111], two with a quantitative or mixed design and one with a qualitative design. Four of the 73 citations did not provide definitions of either of the concepts. Among the studies with quantitative and mixed methods, all three concepts were assessed together in 27 reports. Among these 55 studies, all three concepts were defined and assessed in one citation [84], and no concepts were defined or assessed in two citations [87,88]. A definition for all three concepts was only identified in n = 2 of the included reports [84,111]. The two concepts most often defined in the same report were health literacy and quality of life (n = 7), followed by health literacy and social inequality in health (n = 6), and quality of life and social inequality in health (n = 1). Among the studies with a quantitative or mixed design, the two concepts most often assessed in the same report were health literacy and quality of life (n = 17), followed by social inequality in health and health literacy (n = 3), and quality of life and social inequality in health (n = 1). Only one citation among the reviews, case, or qualitative studies, or the orientational reports, included in the review defined all three concepts [111], and four citations did not define any of the concepts [100,101,109,113].

### 3.4. Results of the Consultation Exercise

The preliminary findings of the scoping review were presented at a multidisciplinary national conference comprising researchers interested in health literacy research and who held positions in clinical care, higher education, and academia, as well as government positions. The preliminary findings were presented at research group meetings. A presentation of the preliminary findings has generated discussion among the members of our research group, with one concern being the paucity of studies on children and on mental health.

## 4. Discussion

### Principal Findings and Comparison with Prior Work

The existing literature on the combined study of social inequality in health, health literacy, and quality of life is characterized by heterogeneity in geography, contexts, populations, and research designs. The degree to which the researchers stated the definitions and methods used to assess the concepts also varied considerably. It seems that the study of social inequality in health, health literacy, and quality of life is often expressed through an investigation of health literacy and quality of life where social inequality appears as the contextualizing factor.

Although health literacy seems to be the most frequently defined concept, social inequality in health is the most often measured, though it is rarely defined among the citations in this review. The association between social inequality in health and health literacy appears to be more applied in the selected publications than the association between either of the two and quality of life. This tendency is in line with previous research, showing a relationship between social inequality in health and health literacy [3,4], and also suggesting associations between health literacy and quality of life [24].

Of utmost importance to public health, the United Nations’ Sustainability Goals [6] clearly define global goals for a sustainable future. Public health promotion is a cornerstone of the United Nations’ Sustainability Goals, and, to achieve these, future research and policy makers should address all three concepts included in the current scoping—inequality in health, health literacy, and quality of life—supporting the rationale of the present review. The relevance of inequality in health, health literacy, and quality of life on the sustainability goals depends on several circumstances, for instance on geographical area, the nature of the population, and level of governance (context) of a study, and they will assert themselves differently because researchers place different levels of emphasis on these areas. For example, the goal of “good health and well-being” is equally important around the world, but the perception of what is good health will be context dependent. Similarly, the extent of social inequality in health will vary, and the variables determining a social gradient will vary. Although we found the highest number of studies from North America, a reasonable number of studies from Asia, South America, and Africa were identified. Based on the findings from this review, we argue that the three concepts are of interest globally—either independently or in combination—because they appear related to each other [1,3,24]. However, acknowledging the considerable differences in countries, both economically and culturally, and regarding the organization of and access to health care systems, future research should develop accordingly in different societal backgrounds to capture the extent of inequalities. In line with the United Nations’ Sustainability Goals [6], future research should pursue investigations considering public health services within different geographical contexts as one major area of potential to explore relevant interventions aiming to target all the three terms at hand.

Our scoping review serves to identify areas that are studied less extensively and that may deserve more attention in the future. We have identified a relative paucity of studies on children and of studies on mental health issues. Children hold great potential for interventions aiming at health promotion and reduced inequalities, either through the children themselves, or, for instance, in a school setting, through their parents, or through more societal interventions. Thus, we anticipated a higher number of studies on this population. Similarly, our review has identified few reports from populations at risk or already suffering from mental health problems, despite the increasing level of mental health problems globally [114]. We would anticipate more research on mental health problems because people with mental health problems frequently have lower levels of both health literacy and quality of life [115,116]. Mental health issues do not always follow a social gradient, but there is a higher proportion of people with mental health problems among those with a low income, those unemployed, those in poor neighbourhoods, or those affected by various forms of racism [116]. Thus, future research should pursue an understanding of how health literacy is at play in people with mental health problems. In addition, the generic construct EQ-5D is often applied among populations to assess and compare quality of life; however, the three response levels mainly detect patients with extreme problems, reducing its sensitivity to health changes [117]. Although methodological considerations were not a focus in our review, the importance of choosing the most appropriate measure is particularly important when using EQ-5D because the EQ-5D-3L may dilute the association between quality of life and social inequality in health through socioeconomic variables.

Compared with mental health, most studies identified in the present review were primarily aimed at patient populations with a variety of somatic diseases, many of which were chronic illnesses. Some publications have addressed vulnerable patient groups in low-income or minority/immigrant settings. Explanations for why these populations seem overrepresented may include a long tradition of applying the three study concepts to health research and a perceived greater susceptibility of patients to socioeconomic inequalities in health care or loss of quality of life. Furthermore, one may anticipate that eventual interventions targeting, for instance, health literacy might be more effective in patients already suffering from disease, especially chronic diseases.

We found it challenging to assess to what degree the three concepts of interest—inequality in health, health literacy, and quality of life—have been used and evaluated in combination in the identified citations. A major reason for these difficulties was the heterogeneity or lack of definitions and measurements used. All three concepts can be defined and assessed differently. For instance, a multidimensional understanding of health literacy should be followed by using a multidimensional measure, such as the definition by Sørensen et al. [118], which is associated with the HLS-EU multidimensional measure [34]. This is equally important in research on quality of life [119,120]. When it comes to social inequality in health, a clear rationale for the choice of variables is even more crucial when trying to understand the chosen concept in context, either through a framework or other conceptualization. This is particularly important because inequalities can occur along socioeconomic, political, ethnic, and cultural axes [8], so the relevant variables are numerous. In the reviewed citations, we found the lowest number of definitions and highest number of measurements for social inequality in health, often through the use of only one determinant, such as race or income. This may be explained by the nature of the concepts, where health literacy and quality of life are easier to define and measure, as well as their multidimensionality. Social inequality in health refers more directly to the basic economic, social, and cultural structures than to the concepts of health literacy and quality of life. Among our included reports, Webb [51] describes the impact of the social determinants of health, health literacy, and quality of life through a case study of a two-year-old child with sickle cell disease, thoroughly presenting all three concepts alongside the available screening tools to assess them. Through the perspective of this child, valuable insights are generated pertaining to patients receiving transfusions on a regular basis. In addition, Walker et al. [92] describe how the social determinants of health include the social and economic conditions that influence health status. Both of these latter studies constitute examples for future reference.

To reduce social inequality in health, efforts must be put into political and structural change and economic reallocation. The realization of this goal demands a broad perspective rather than projects aiming to enhance one concept at a time without acknowledging other possibly relevant factors. Here, a high number of cross-sectional studies is expected but may add less valuable information to the understanding of these relationships than the few studies with longitudinal or even interventional designs [19]. Seemingly, the field is still more concerned with descriptive research. One explanation for this is that the research on the relationship between the three concepts is still in its infancy, which is in line with the rationale of our scoping review. Although causal, mediating, or moderating relationships are suggested for two and two of our three concepts, more research is needed to establish the associations between all three.

## 5. Limitations

Although we have aimed to conduct a thorough and systematic scoping search in multiple databases, there is the risk of us having missed relevant research of value for our review. Hand searches were not performed, and grey literature was not included. There exists a substantial amount of grey literature on this subject that was not included because our review was limited to the inclusion of studies published in peer-reviewed journals. Our three concepts were used differently in the identified literature, and, most clearly, this was a challenge for uniformly evaluating social inequality in health. Social determinants of health and social inequality in health are used interchangeably [121]. Thus, our systematic search included both. Although we had a clear purpose and understanding of the concepts beforehand, the authors may have had other intentions, causing possible inconsistencies between our aims and theirs. On the other hand, the different perspectives from the members of our research group, all with diverse backgrounds and research interests, allowed for a broad a priori understanding of the concepts, strengthening our study. We did not conduct a concept analysis according to methods for terminology work, although we recognize that the construction of controlled vocabularies in specialized contexts might affect the terms we included in this scoping review. Future research should investigate this more in detail, alongside the intended definitions of concepts that vary a great deal. Lastly, we did not conduct an appraisal of the methodological quality of the included studies, which is in line with the scoping review method [32], thus, we have not addressed any risk of bias in or between the included studies.

## 6. Conclusions

The present scoping review reveals a complex and evolving field of research that integrates inequality in health, health literacy, and quality of life. The included papers report on a geographical spread, biased toward North America, and on a heterogeneous range of populations and contexts. Various patient groups were represented, with a particular focus on patient groups suffering from chronic illnesses. Notably, the included reports tend to measure inequality in health, health literacy, and quality of life more than they define them, underscoring the emergent nature of this interdisciplinary field. Given the novel state of research that intertwines these concepts, it may perhaps be too early to discern their fully entangled association. To better inform public health policy decisions and contribute to the United Nations’ Sustainability Goals, future research should prioritize a clear definition and robust integration of these three public health indicators—as well as an in-depth exploration of their interrelationships.

## Figures and Tables

**Figure 1 ijerph-21-00036-f001:**
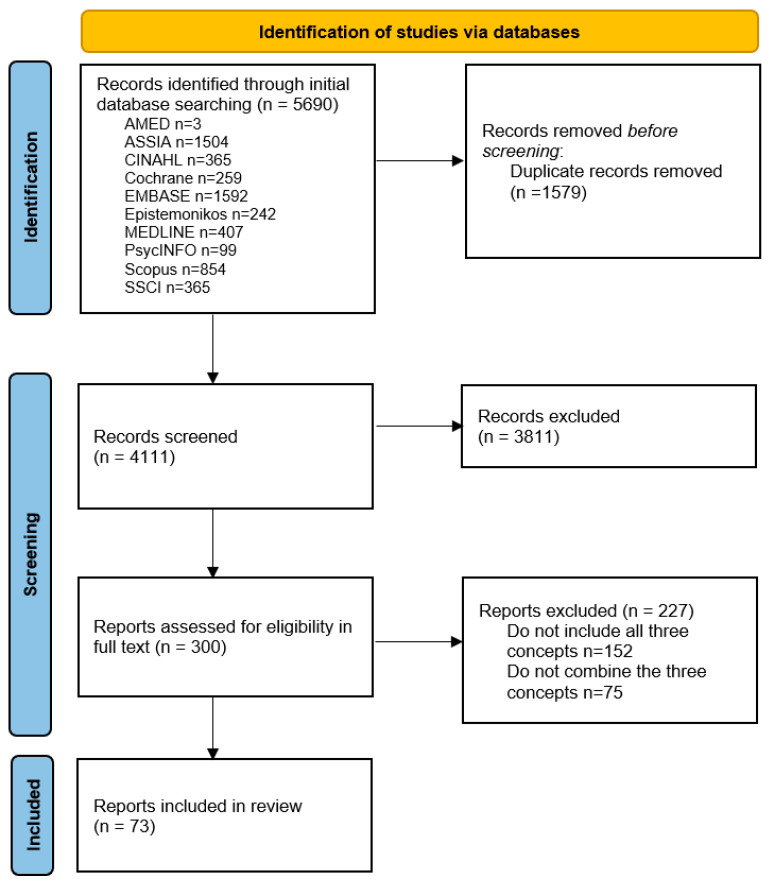
Flow chart.

**Figure 2 ijerph-21-00036-f002:**
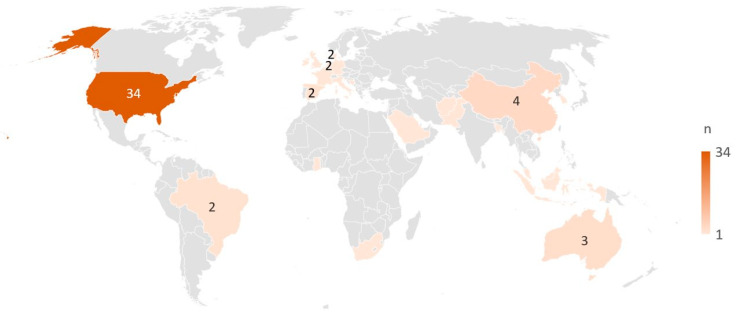
Countries represented by the included primary studies (n = 61). All countries with ≥2 studies are marked with the specific number, while countries with only one study are marked by colour only.

**Figure 3 ijerph-21-00036-f003:**
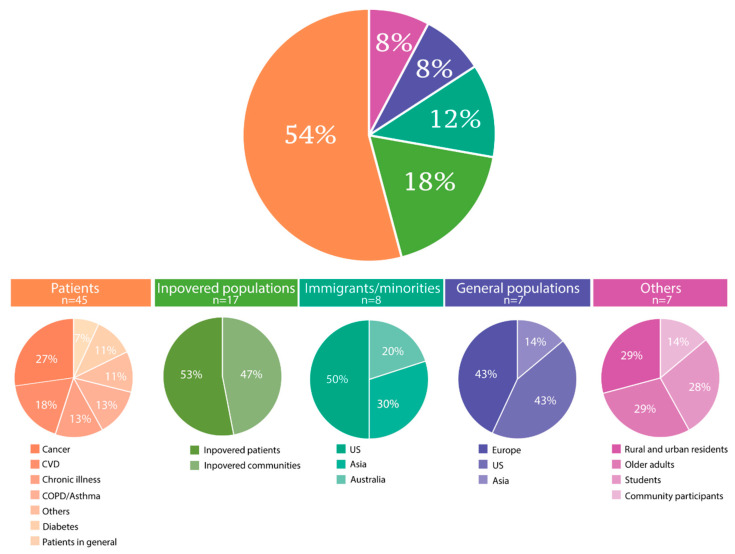
Categorization of study populations as covered by the included citations (N = 73). Twelve studies reported on a population in two categories.

**Figure 4 ijerph-21-00036-f004:**
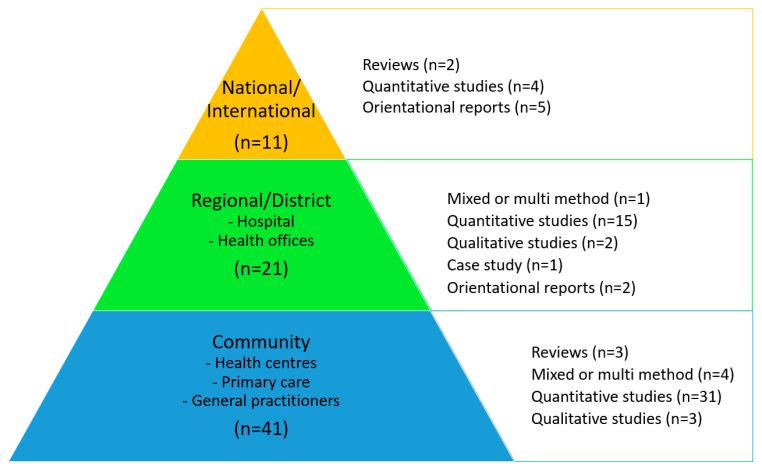
Contextual levels and research designs of the 73 included citations. Orientational reports include statements, editorials, and consensus papers.

**Table 1 ijerph-21-00036-t001:** Population, concept, and context tool to set eligibility criteria.

PCC Element	Scoping Review Target	Inclusion Criteria	Exclusion Criteria
**Population**	Citations reflecting on or involving humans throughout a life course.	All ages and all life situations reflecting or involving humans from birth to death.	Animal studies. Autopsy studies.
**Concept**	Any associations between the concepts of social inequality in health, health literacy, and quality of life.	All three concepts must be applied in their widest form together in the research citation (according to the search terms).	Citations reporting on only one or two of the concepts.
**Context**	All contexts involving humans.		

**Table 2 ijerph-21-00036-t002:** (**a**) Definitions * and assessments * among the quantitative or mixed methods primary studies included in the review n = 55. (**b**) Definitions * among the reviews, case, or qualitative studies, and the orientational reports included in the review, n = 18.

(a)						
Author	Social Inequality in Health Definition	Social Inequality in Health Assessment	Health Literacy Definition	Health Literacy Assessment	Quality of Life Definition	Quality of Life Assessment
Alruthia, Y., et al. [52]						
Amoah, P. A., et al. [53]						
An, J. Y., et al. [54]						
Angner, E., et al. [55]						
Apter, A. J., et al. [56]						
Apter, A. J., et al. [57]						
Asare, M., et al. [58]						
Batista, M. J., et al. [59]						
Blancafort Alias, S., et al. [60]						
Clarke, H., et al. [61]						
Curtis, L. M., et al. [62]						
Durand, M. A., et al. [63]						
Ernsting, C., et al. [64]						
Faruqi, N., et al. [65]						
Fung, C. S. C., et al. [47]						
Goss, H. R., et al. [66]						
Graham, L. A., et al. [67]						
Guhl, E., et al. [68]						
Harsch, S., et al. [69]						
Hickey, K. T., et al. [70]						
Irwin, K., et al. [44]						
Jamieson, L. M., et al. [41]						
Johnson, D. R., et al. [71]						
Katzmarzyk, P. T., et al. [72]						
Kim, S. P., et al. [73]						
Kim, S. P., et al. [74]						
Lang, L. P. [75]						
Langton, C. E. [49]						
Macabasco-O’Connell, A. et al. [76]						
McDougall, J. A., et al. [77]						
Meyers, A. G., et al. [78]						
Miller, D. B., et al. [79]						
Myaskovsky, L., et al. [80]						
Omachi, T. A. et al. [81]						
Ownby, R. L., et al. [82]						
Park, N. H., et al. [83]						
Prihanto, J. B., et al. [84]						
Rak, E. C. [85]						
Reid, A. L., et al. [50]						
Rijken, M., et al. [86]						
Roberto, L. L., et al. [87]						
Scheuer, S. [88]						
Simon, M. A., et al. [89]						
Tan, S. S., et al. [42]						
Todorovic, N., et al. [90]						
Uddin, M. N., et al. [43]						
Virlée, J., et al. [91]						
Walker, R. J., et al. [92]						
Wang, C., et al. [93]						
Wang, C., et al. [94]						
Washington, D. M., et al. [46]						
Wong Min, F., et al. [45]						
Xiao, Z., et al. [95]						
Xu, R. H., et al. [96]						
Aaby, A., et al. [97]						
(**b**)						
**Author**	**Social Inequality in Health Definition**		**Health Literacy Definition**		**Quality of Life Definition**	
**Reviews**						
Ghisi, G., et al. [98]						
Gibbs, J. F., et al. [99]						
Maliski, S. L., et al. [100]						
Schaffler, J., et al. [101]						
Stormacq, C., et al. [102]						
**Qualitative Studies**						
Hardgraves, V. M., et al. [103]						
Ladak, L. A., et al. [48]						
Lowe, S. M., et al. [104]						
Talmage, C. A., [105]						
White, B. M., [106]						
**Case Study**						
Webb, J. [51]						
**Orientational Reports**						
Albus, C., [107]						
Bragard, L., et al. [108]						
de Vries, E., et al. [109]						
Griech, S. F., et al. [110]						
Kuehnert, et al. [111]						
Merriman, B., et al. [112]						
Rozier, R. G. [113]						

* Grey areas indicate the presence of definitions and assessments. * Grey areas indicate the presence of definitions.

## Data Availability

All search results can be made available upon request to the first author.

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
