# Peer review of "Unpacking the Public Health Triad of Social Inequality in Health, Health Literacy, and Quality of Life—A Scoping Review of Research Characteristics"

_ijerph, 2023, doi:10.3390/ijerph21010036_

Round 1
Reviewer 1 Report
Comments and Suggestions for Authors
the following title is recommended to use:
Unveiling the Public Health Triad: Exploring Social Inequality in Health, Health Literacy, and Quality of Life - A Comprehensive Scoping Review.
Introduction:
The background provided lacks clarity and conciseness. The information is presented in a lengthy and convoluted manner, making it difficult for readers to grasp the main points. Additionally, the background fails to clearly state the objective of the study and the significance of exploring the interrelationships between social inequality in health, health literacy, and quality of life. the necessity of studying life style should be highlighted. use the following references
Siahpoosh, Marzieh Beygom, and Soheil Gholibeygi. "Exercise in Winter: A Hypothesis from Persian Medicine to Improve Healthy Lifestyle." International Journal of Sport Studies for Health 6.1 (2023).
Methodes
The methods described in the passage seem to be well-structured and comprehensive for conducting a scoping review. However, there are a few potential areas for criticism:
1. Lack of justification: The passage does not provide a clear rationale for why a scoping review was chosen as the methodology. It would have been helpful to explain why a scoping review was deemed appropriate and how it differs from other review methods.
2. Search strategy: While the passage mentions the use of various databases for the literature search, it does not provide details about the specific search terms or search strings used. This lack of transparency makes it difficult to evaluate the comprehensiveness of the search.
3. Inclusion and exclusion criteria: The passage briefly mentions the eligibility criteria for selecting studies but does not provide a clear justification for the specific criteria chosen. It would have been beneficial to explain the reasoning behind the inclusion and exclusion criteria to ensure the selection process is unbiased.
4. Data extraction: The passage mentions the use of a charting form for data extraction, but it does not provide details about the specific data items extracted or the process used for data synthesis. This lack of information makes it challenging to assess the reliability and validity of the data extraction process.
Results
Good
Discussion
One criticism of this discussion is that it lacks a clear and concise thesis statement. The author provides a lot of information about the existing literature on social inequality in health, health literacy, and quality of life, but it is not clear what their main argument or point is. This lack of focus makes it difficult for the reader to understand the purpose and significance of the discussion. Additionally, the discussion relies heavily on citations and references to other studies without providing much original analysis or interpretation. While it is important to acknowledge the existing literature and build upon previous research, the author should also provide their own insights and contribute to the ongoing conversation on the topic.
Furthermore, the discussion lacks a clear structure or organization. The ideas and arguments presented are scattered and do not flow logically from one point to another. This makes it challenging for the reader to follow the author's train of thought and understand the overall message of the discussion.
Author Response
Dear reviewers and assistant editor
Thank you for the thorough and insightful comments to our manuscript on Social Inequality in Health, Health Literacy, and Quality of Life. We have reflected on these comments and provide a point-by-point answer in the following text, with consecutive revisions in the manuscript. We believe that the remarkable point made by the reviewers have significantly improved our manuscript, and we are humble by the efforts made by the reviewers to assess our research report.
On behalf of the authors,
Heidi Holmen
------------------------------------------------------------------
Reviewer 1
the following title is recommended to use:
Unveiling the Public Health Triad: Exploring Social Inequality in Health, Health Literacy, and Quality of Life - A Comprehensive Scoping Review.
Response from authors: We are thankful for the suggested heading provided by the reviewer, and we are glad it seems to maintain most of our initial meaning with our heading. We would like to argument for keeping the heading as it is now, and first, our review aims to unpack more than it unveils, as the field of Social Inequality in Health, Health Literacy, and Quality of Life consists of many bits and pieces, nor previously characterized in relation to each other, thus, we would like to keep “unpacking” in our title. Second, and again referring to the field consisting of many bits and pieces that we are characterizing, we would like to keep the title’s second part as “a scoping review of research characteristics” to ensure the reader of the report does not expect an in-depth analysis as one would expect in a meta-analysis og meta-synthesis. We hope the reviewer can recognize the benefits of keeping the title as it is, and that it better suits the content of the report.
Introduction: The background provided lacks clarity and conciseness. The information is presented in a lengthy and convoluted manner, making it difficult for readers to grasp the main points. Additionally, the background fails to clearly state the objective of the study and the significance of exploring the interrelationships between social inequality in health, health literacy, and quality of life. the necessity of studying life style should be highlighted. use the following reference: Siahpoosh, Marzieh Beygom, and Soheil Gholibeygi. "Exercise in Winter: A Hypothesis from Persian Medicine to Improve Healthy Lifestyle." International Journal of Sport Studies for Health 6.1 (2023).
Response from authors: We thank the reviewer for pointing to the utmost importance of lifestyle when discussing Social Inequality in Health, Health Literacy, and Quality of Life in a public health perspective. The intention of the review we have conducted is however to try to investigate Social Inequality in Health, Health Literacy, and Quality of Life as more underlying aspects affecting the possibilities to undertake more or less healthy lifestyle choices. This way, we acknowledge the point made by the reviewer in the fact that lifestyle is of importance when studying all the three chosen domains. Furthermore, as we value the suggested reference and have read it with great interest, we cannot see how this primary study would pin-point our introduction better. Also, we have strived to present the introduction as broad as necessary but as short as possible. The first section introduces the three terms together, whilst we have two sections for the terms at hand to more in depth define them properly, which, consistent with the findings of our review is what most often is lacking in the published research. In addition, we must present how they already are found to be intertwined before the knowledge gap is presented. Thus, our introduction briefly places our study in a broad context and defines the key concepts together with cited key publications, as requested by the journal. We have clarified the aims more, and we do hope it reads better now. “The following primary research question guided this study:
- What are the characteristics of the studies that use the three concepts’ social inequality in health, health literacy and quality of life together?
- How are the three concepts presented in the research?
- Which definitions and measurements are used for the three concepts?”
Methods: The methods described in the passage seem to be well-structured and comprehensive for conducting a scoping review. However, there are a few potential areas for criticism:
- Lack of justification: The passage does not provide a clear rationale for why a scoping review was chosen as the methodology. It would have been helpful to explain why a scoping review was deemed appropriate and how it differs from other review methods.
Response from authors: We thank the reviewer for being cognizant on the rationale of scoping reviews. We understand that the original text considering how a full review was considered premature, and we have added a sentence to clarify, and we sincerely hope the reviewer finds a scoping review justified: “To better inform future systematic reviews, meta-syntheses or meta-analyses, a scoping review was seen as crucial to investigate the characteristics of this field, to ensure that future reviews extract and synthesize relevant variables in the required contexts and populations.”
- Search strategy: While the passage mentions the use of various databases for the literature search, it does not provide details about the specific search terms or search strings used. This lack of transparency makes it difficult to evaluate the comprehensiveness of the search.
Response from authors: We are most thankful for this important comment on the transparency of our review. While we had the searches in the supplementary file uploaded and referred to at the end of the manuscript, unfortunately this reference was not in the text in the section on the search. We have now added the following sentence on page 7, top section: “(…), and all search strings are found in the Supplementary file 2.”
- Inclusion and exclusion criteria: The passage briefly mentions the eligibility criteria for selecting studies but does not provide a clear justification for the specific criteria chosen. It would have been beneficial to explain the reasoning behind the inclusion and exclusion criteria to ensure the selection process is unbiased.
Response from authors: We thank the reviewer for raising the concern of un unbiased selection process. In the manuscript, we refer to a research question considering the terms of interest, repeated in Table 1, outlining the three terms in target together with a consequent inclusion and exclusion criteria, and further specified in the text on identification of studies with examples on how the inclusion and exclusion criteria was interpreted in the research team. Altogether, this extensive presentation on the eligibility is given to reduce the risk of bias in the data selection, and we sincerely hope the reviewer would assess the necessity of adding even more details as we are afraid it might be length and preclude the readability of our text. Please, we welcome any advice on how to specify this even better.
- Data extraction: The passage mentions the use of a charting form for data extraction, but it does not provide details about the specific data items extracted or the process used for data synthesis. This lack of information makes it challenging to assess the reliability and validity of the data extraction process.
Response from authors: We are thankful for the thorough read made by the reviewer, and have, as requested, added details on the data extracted through a mention of all points extracted. We do hope this provides the necessary level of detail for the reader to assess our work. Bottom of page 7: “Data on the definition and use of the three terms included 1. the relevant definition for all three concepts as written in the paper, if any, 2. any use (and definitions) of interchangeable concepts to either of the three, 3. A note on where in the paper the concepts were applied, 4. A note on how the three concepts are combined, 5. A note on which, if any is seen as the main concept and whether there is a hierarchical relationship between the concepts, and 6. Summary note on the included and extracted report.”
Results: Good
Response from authors: Thank you for acknowledging the quality and readability of our results section.
Discussion: One criticism of this discussion is that it lacks a clear and concise thesis statement. The author provides a lot of information about the existing literature on social inequality in health, health literacy, and quality of life, but it is not clear what their main argument or point is. This lack of focus makes it difficult for the reader to understand the purpose and significance of the discussion. Additionally, the discussion relies heavily on citations and references to other studies without providing much original analysis or interpretation. While it is important to acknowledge the existing literature and build upon previous research, the author should also provide their own insights and contribute to the ongoing conversation on the topic.
Response from authors: We thank the reviewer for being critical to the readability of our manuscript. We have added a section heading in the discussion “Principal findings and comparison with prior research”. Our aim was to identify and describe the characteristics of research that intersects social inequality in health, health literacy and quality of life. This way, the discussion of consequently ordered based on the results of this scoping review, that is, discussing the principal findings in light of previous research. The first section summarises the main finding, before we discuss section by section, our findings with our arguments and interpretations and whether they are supported or not in previous literature. Where there is no such literature, we clarify the need for more research. We draw on the SDG by the UN to clarify the relevance of our work, and we identify clear lacks of research, such as on children. Although we are not entirely sure how to meet the requirements of the reviewer, we have slightly reviewed the discussion, also in line with the points of reviewer 2, and we sincerely hope it reads better for both reviewers.
Furthermore, the discussion lacks a clear structure or organization. The ideas and arguments presented are scattered and do not flow logically from one point to another. This makes it challenging for the reader to follow the author's train of thought and understand the overall message of the discussion.
Response from authors: see the above response.

Reviewer 2 Report
Comments and Suggestions for Authors
Heidi Holmen et al. submitted to IJERPH a scoping review, focusing to the public health triad, focusing to the social inequality in health, health literacy and quality of life.
This manuscript is well structured and very interesting and lends itself to numerous insights, as it is a very vast topic.
Therefore, I propose below my suggestions for the improvement:
- It would be very useful performing discussions in critical thinking, examining in depth if Public Health Services can contribute to reducing inequalities by ensuring mandatory vaccinations and reaching all eligible subjects; guarantee meals of adequate nutritional and hygienic quality in schools; ensuring controls on the quality of water intended for human consumption; monitoring environmental contamination parameters; guaranteeing controls on workplace safety in the various supply chains or sectors. These topics are very interesting for readers professionally involved in Preventive Medicine Services and, if well developed, can give an added value to discussions of this important manuscript. Thank you.
- There are some typos, different writing styles used, references should appear in the text as indicated in the Instructions for Authors (e.g. in square brackets) and references must be indicated as expected by IJERPH.
Comments on the Quality of English LanguageMinor editing of English language required.
Author Response
Dear reviewers and assistant editor
Thank you for the thorough and insightful comments to our manuscript on Social Inequality in Health, Health Literacy, and Quality of Life. We have reflected on these comments and provide a point-by-point answer in the following text, with consecutive revisions in the manuscript. We believe that the remarkable point made by the reviewers have significantly improved our manuscript, and we are humble by the efforts made by the reviewers to assess our research report.
On behalf of the authors,
Heidi Holmen
Reviewer 2
Heidi Holmen et al. submitted to IJERPH a scoping review, focusing to the public health triad, focusing to the social inequality in health, health literacy and quality of life. This manuscript is well structured and very interesting and lends itself to numerous insights, as it is a very vast topic.
Response from authors: Thank you for acknowledging the quality and readability of our work, and the relevance of its conduct.
Therefore, I propose below my suggestions for the improvement:
- It would be very useful performing discussions in critical thinking, examining in depth if Public Health Services can contribute to reducing inequalities by ensuring mandatory vaccinations and reaching all eligible subjects; guarantee meals of adequate nutritional and hygienic quality in schools; ensuring controls on the quality of water intended for human consumption; monitoring environmental contamination parameters; guaranteeing controls on workplace safety in the various supply chains or sectors. These topics are very interesting for readers professionally involved in Preventive Medicine Services and, if well developed, can give an added value to discussions of this important manuscript. Thank you.
Response from authors: We appreciate this comment by the reviewer and find it intriguing. We have strengthened the focus on public health in the discussion and have added some details according to the suggestions made by the reviewer. We have made some suggestions in line with which populations future research should include, but not so much the interventions, as they could be many, but including public health services is very much relevant. Thank you.
Revised text, page 17 “Of utmost importance to public health, the United Nations’ Sustainability Goals [114] clearly define global goals for a sustainable future. Public health promotion is a cornerstone of the United Nations’ Sustainability Goals, and to achieve these, future research and policy makers should address all three concepts included in the current scoping – inequality in health, health literacy, and quality of life – supporting the rationale of the present review. The relevance of inequality in health, health literacy, and quality of life on the sustainability goals depends on several circumstances, for instance on geographical area, the nature of the population and level of governance (context) of a study, and they will assert themselves differently because researchers place different levels of emphasis on these areas.”
Revised text, page 18 “In line with the United Nations’ Sustainability Goals [114], future research should pursue investigations considering public health services within different geographical contexts as one major area of potential to explore relevant interventions aiming to target all three terms at hand.”
- There are some typos, different writing styles used, references should appear in the text as indicated in the Instructions for Authors (e.g. in square brackets) and references must be indicated as expected by IJERPH.
Response from authors: We thank the reviewer for the thorough read and apologize for the lack of adherence to the manuscript style, which we have strived to adhere to in the revised manuscript. We are most thankful if the reviewer would advise us on further typos if we have not been able to detect them all.